# Ceapins block the unfolded protein response sensor ATF6α by inducing a neomorphic inter-organelle tether

Sandra Elizabeth Torres[1,2,3], Ciara M Gallagher[2,3†‡], Lars Plate[4,5†], Meghna Gupta[2], Christina R Liem[1,3], Xiaoyan Guo[6,7], Ruilin Tian[6,7], Robert M Stroud[2], Martin Kampmann[6,7], Jonathan S Weissman[1,3*], Peter Walter[2,3*]

[1]Department of Cellular and Molecular Pharmacology, University of California, San Francisco, San Francisco, United States; [2]Department of Biochemistry and Biophysics, University of California, San Francisco, San Francisco, United States; [3]Howard Hughes Medical Institute, University of California, San Francisco, San Francisco, United States; [4]Department of Chemistry, Vanderbilt University, Nashville, United States; [5]Department of Biological Sciences, Vanderbilt University, Nashville, United States; [6]Department of Biochemistry and Biophysics, Institute for Neurodegenerative Diseases, University of California, San Francisco, San Francisco, United States; [7]Chan Zuckerberg Biohub, San Francisco, United States

*For correspondence:
Jonathan.Weissman@ucsf.edu
(JSW);
peter@walterlab.ucsf.edu (PW)

†These authors contributed
equally to this work

Present address: ‡Cairn
Biosciences, San Francisco,
United States

Competing interests: The
authors declare that no
competing interests exist.

Reviewing editor: Elizabeth A
Miller, MRC Laboratory of
Molecular Biology, United
Kingdom

**Abstract** The unfolded protein response (UPR) detects and restores deficits in the endoplasmic reticulum (ER) protein folding capacity. Ceapins specifically inhibit the UPR sensor ATF6α, an ER-tethered transcription factor, by retaining it at the ER through an unknown mechanism. Our genome-wide CRISPR interference (CRISPRi) screen reveals that Ceapins function is completely dependent on the ABCD3 peroxisomal transporter. Proteomics studies establish that ABCD3 physically associates with ER-resident ATF6α in cells and in vitro in a Ceapin-dependent manner. Ceapins induce the neomorphic association of ER and peroxisomes by directly tethering the cytosolic domain of ATF6α to ABCD3's transmembrane regions without inhibiting or depending on ABCD3 transporter activity. Thus, our studies reveal that Ceapins function by chemical-induced misdirection which explains their remarkable specificity and opens up new mechanistic routes for drug development and synthetic biology.
DOI: https://doi.org/10.7554/eLife.46595.001

## Introduction

The endoplasmic reticulum (ER) is the site of folding and assembly of secreted and transmembrane proteins. When ER homeostasis is disturbed, misfolded proteins accumulate and activate the unfolded protein response (UPR) (*Walter and Ron, 2011*). One of the ER-resident UPR sensors, ATF6α, is an ER-tethered transcription factor that is cytoprotective and necessary for cell survival when cells experience ER stress (*Wu et al., 2007*; *Yamamoto et al., 2007*). Under stress conditions, ATF6α traffics to the Golgi apparatus, where it undergoes intramembrane proteolysis, releasing a bZIP transcription factor domain that moves to the nucleus and activates transcription (*Haze et al., 1999*; *Yoshida et al., 1998*). The events leading to ATF6α activation and trafficking remain poorly understood, but require the Golgi-resident proteases S1P and S2P and general components involved in COPII trafficking (*Nadanaka et al., 2004*; *Okada et al., 2003*; *Schindler and Schekman, 2009*; *Ye et al., 2000*) that are not specific to ATF6α.

Using a cell-based high-throughput screen, we recently identified a series of selective small-molecule inhibitors of ATF6α signaling, termed Ceapins (from the Irish verb 'ceap' meaning 'to trap') (*Gallagher et al., 2016*). Ceapins act on the most upstream step of ATF6α activation by retaining ATF6α at the ER and excluding it from ER exit sites during ER stress. When this trafficking requirement is removed by collapsing the Golgi apparatus into the ER, making ATF6α accessible to S1P and S2P, ATF6α is still cleavable by the proteases in the presence of Ceapin. Upon Ceapin treatment, ATF6α rapidly and reversibly forms foci without requiring new protein synthesis (*Gallagher et al., 2016*; *Gallagher and Walter, 2016*). The molecular target(s) of Ceapins, let alone how Ceapins specifically inhibit ATF6α, especially in light of the fact that activation depends on components that are shared by other cellular process, have remained an enigma.

To identify the molecular target of Ceapin, we carried out an unbiased genome-wide screen and proteomic analysis. Our approaches converged on a single target, the peroxisomal transporter ABCD3. ATF6α and ABCD3 normally do not interact and, indeed, localize to different parts of the cell. Ceapins induce these novel physical associations between ATF6α and ABCD3 in cells and in vitro. Our results indicate that Ceapins achieve their remarkable specificity through an unprecedented mechanism of small molecule induced inter-organelle tethering.

## Results

### ABCD3 KD desensitizes cells to Ceapin-A7

To decipher the molecular mechanism of Ceapins, we carried out a genome-wide CRISPR interference (CRISPRi) screen to identify genes whose knockdown (KD) resulted in reduced or enhanced sensitivity to the drug. To this end, we screened a genome-wide sgRNA library (*Horlbeck et al., 2016*) in K562 cells that stably expressed dCas9-KRAB and an mCherry transcriptional reporter dependent on ATF6α activation (*Figure 1A*). Treatment with tunicamycin (Tm), which blocks N-linked glycosylation, activates ATF6α signaling leading to a two-fold reporter induction that was completely dependent on ATF6α (*Figure 1A*). As a positive control, knocking down *MBTPS2*, one of the Golgi proteases that processes ATF6α, also inhibited induction of the reporter (*Figure 1—figure supplement 1A*), whereas knocking down *HSPA5*, encoding the major Hsp70-type ER chaperone BiP (Binding Protein), induced ER stress and the reporter constitutively (*Figure 1—figure supplement 1B*).

To carry out our genome-wide screen, we transduced the K562 ATF6α reporter cell line and selected for sgRNA expressing cells. We then induced ER stress with Tm in the presence or absence of Ceapin-A7, a potent member of the Ceapin family, and sorted cells by FACS (fluorescence-activated cell sorting). We isolated populations with decreased or increased ATF6α signaling (bottom 30% and top 30% of the reporter signal distributions, respectively) and used next-generation sequencing to quantify frequencies of cells expressing each sgRNA in both pools to evaluate how expression of each individual sgRNA affects activation of the ATF6α reporter (*Adamson et al., 2016*; *Sidrauski et al., 2015*) (*Figure 1B*).

As expected, KD of *ATF6α* or *MBTPS2* (encoding S2P) inhibited reporter induction (*Figure 1C*). Knocking down abundant ER quality control components such as *HSPA5*, induced ER stress and turned on the reporter independently of Ceapin treatment (labeled in *red* in *Figure 1C*, *Figure 1—figure supplement 1C–D*). Ceapin independent genes localized to the diagonal because their knockdown changed the expression of the reporter to the same degree in both treatments (labeled in red in *Figure 1—figure supplement 1C*). Of particular interest were genes whose KD specifically made cells insensitive to Ceapin treatment allowing activation of the reporter by Tm in the presence of Ceapin (labeled in black in *Figure 1—figure supplement 1C*). Two genes, *ABCD3* and *PEX19*, robustly retested among the more than twenty hits from the genetic screen we individually knocked down and tested in the ERSE reporter cell line.

ABCD3, which encodes a peroxisomal ABC transporter involved in long-chain fatty acid import into peroxisomes, desensitized cells to Ceapin treatment (*Figure 1C*, *Figure 1—figure supplement 1C–D*). Additionally, *PEX19*, which is necessary for chaperoning and targeting ABCD3 to the peroxisome, also desensitized cells to Ceapin treatment (*Figure 1C*, *Figure 1—figure supplement 1C–D*). We knocked down these candidates individually and performed ERSE-mCherry dose response assays using Tm. Retesting of these candidates revealed that *ABCD3* and *PEX19* KD cells remained

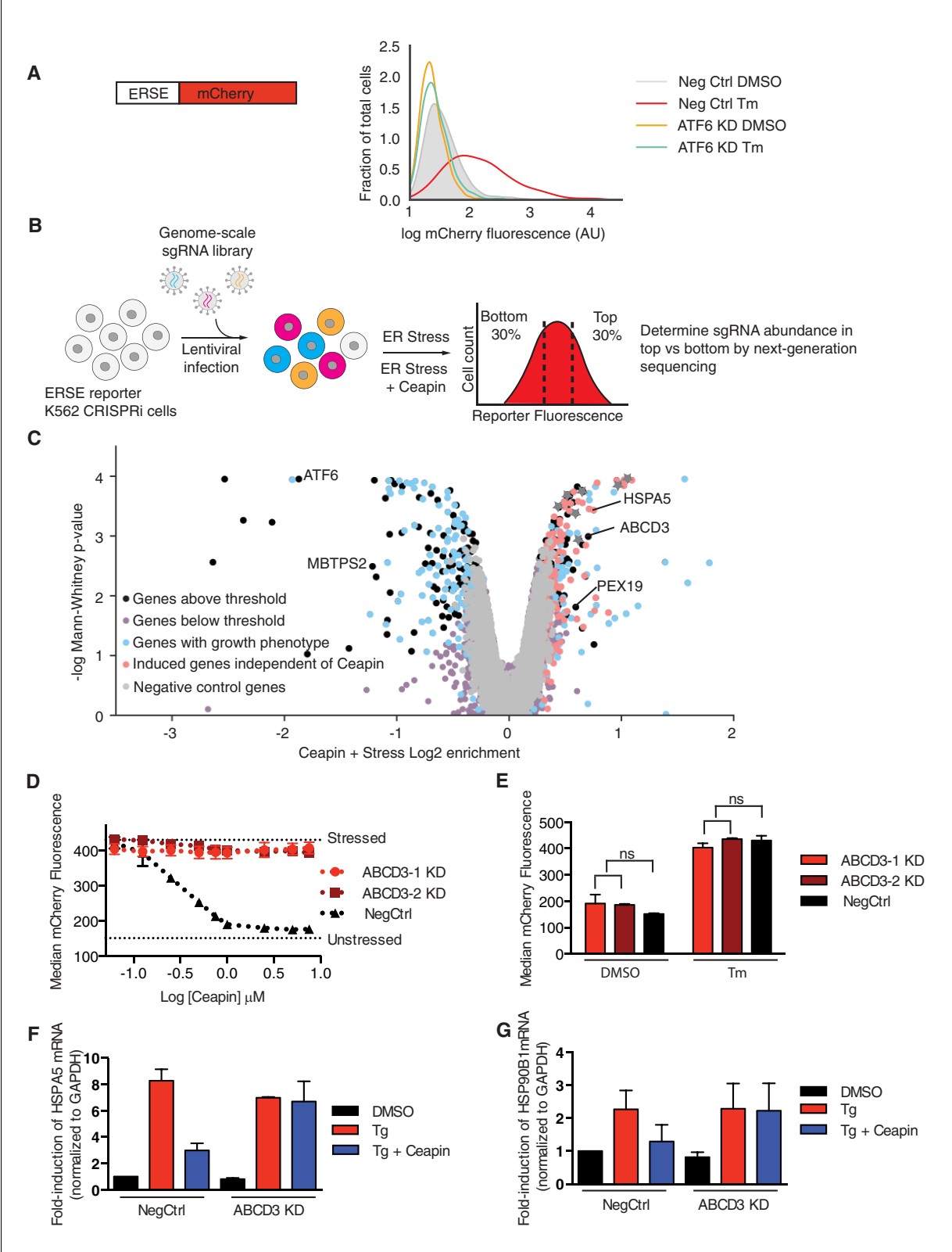

**Figure 1.** ABCD3 KD desensitizes cells to Ceapin-A7. (**A**) Schematic of the ER stress element (ERSE) reporter cassette. K562 ERSE reporter cells were transduced with the indicated sgRNAs and treated with vehicle (DMSO) or tunicamycin (Tm) (6 µg/ml) for 16 hr. (**B**) Schematic of the CRISPRi screen to identify the target of Ceapin. K562 cells expressing the ERSE reporter were transduced with the sgRNA library. The population was then divided into two subpopulations, which were treated with Tm or Tm plus Ceapin-A7 at EC$_{90}$ (3 µM) for 16 hr. Cells in the top and bottom thirds of mCherry

*Figure 1 continued on next page*

*Figure 1 continued*

fluorescence of each subpopulation (Tm-treatment and Tm + Ceapin-treatment) were collected by FACS and processed to measure the frequencies of sgRNAs contained within each. (C) Volcano plot of gene-reporter phenotypes and p values from CRISPRi screen. Negative control sgRNA targeted genes (*gray*), Ceapin-independent genes (*red*), genes with growth phenotypes (*blue*), and Ceapin hits (*black*) are indicated. (*) denotes chromatin architecture and remodeling related genes that impact reporter transcription. The reporter phenotypes and p values for genes in CRISPRi screen are listed in *Figure 1—source data 1*. (D) K562 ERSE reporter cells with individual ABCD3 sgRNAs or control sgRNA (NegCtrl) were treated with Tm and increasing concentrations of Ceapin-A7 for 16 hr. Reporter fluorescence was measured by flow cytometry and median values were plotted (N = 3, ± SD). (E) K562 ERSE reporter *ABCD3* and NegCtrl KD cells were treated with DMSO or Tm and reporter activation was measured as in (D). (F and G) qPCR analysis of ATF6α target genes *HSPA5* and *HSP90B1*, respectively. HepG2 CRISPRi NegCrl and ABCD3 KD cell lines were treated with DMSO, thapsigargin (Tg) (100 nM), and Tg with Ceapin (6 µM). Tg blocks the ER calcium pump and induces ER stress. Data plotted are mRNA levels for *HSPA5* and *HSP90B1* normalized to GAPDH and then compared to unstressed NegCtrl cells ± standard deviation of duplicate technical replicates of two biological replicates.

DOI: https://doi.org/10.7554/eLife.46595.002

The following source data and figure supplements are available for figure 1:

**Source data 1.** Reporter phenotypes and p values for genes in CRISPRi screen.

DOI: https://doi.org/10.7554/eLife.46595.005

**Figure supplement 1.** Genome-scale CRISPRi screen to identify molecular target of Ceapin.

DOI: https://doi.org/10.7554/eLife.46595.003

**Figure supplement 2.** *ABCD3* KD does not affect ATF6α nuclear translocation.

DOI: https://doi.org/10.7554/eLife.46595.004

completely insensitive to Ceapin-A7 at saturating concentrations (*Figure 1D*, *Figure 2—figure supplement 3A*). To determine if ATF6α trafficking, processing, or activation is altered in *ABCD3* KD cells, we then measured ATF6α nuclear translocation (*Figure 1—figure supplement 2*) and the downstream ATF6α-N activation of the reporter and endogenous ATF6α target genes HSPA5 and HSP90B1 (*Figure 1E–G*). In the absence of ER stress, *ABCD3* or *PEX19* KD cells also do not cause constitutive nuclear translocation nor activate ATF6α (*Figure 1E–G*, *Figure 1—figure supplement 2*, *Figure 2—figure supplement 3B*). Furthermore, in the presence of ER stress, *ABCD3* or *PEX19* KD alone did not impede ATF6α nuclear translocation nor activation (*Figure 1E–G*, *Figure 2—figure supplement 3B*). These results indicate that neither ABCD3 nor PEX19 have direct roles in ATF6α signaling, posing the question of how Ceapins functionally connect proteins that reside in separate organelles.

## ABCD3 is required for Ceapin-induced ATF6α foci

Ceapin treatment induces rapid and reversible formation of ATF6α foci that are retained in the ER (*Figure 2A*) (*Gallagher et al., 2016*; *Gallagher and Walter, 2016*). We next tested if ABCD3 was directly involved in the formation of these foci and would colocalize with ATF6α. Indeed, in Ceapin-treated cells, ATF6α colocalized with ABCD3 as visualized by immunofluorescence (*Figure 2A–B*). This result was surprising because newly synthesized ABCD3 is inserted directly into the peroxisomal membrane using PEX19 as import receptor (*Imanaka et al., 1996*; *Biermanns and Gärtner, 2001*; *Kashiwayama et al., 2007*; *Kashiwayama et al., 2005*; *Sacksteder et al., 2000*). ABCD3 is not co-translationally translocated into the ER, indicating there is not a pool of ABCD3 in the ER (*Figure 2—figure supplement 1*) (*Jan et al., 2014*); indeed, it is commonly used as a reliable marker for peroxisomes (*Uhlén et al., 2015*). Since both ABCD3 and PEX19 scored as hits in our screen, it seemed plausible that Ceapin induces ATF6α colocalization with peroxisomal ABCD3. We next tested whether ATF6α also colocalized with other peroxisomal markers, peroxisomal membrane protein PEX14 and peroxisomal matrix protein Thiolase (a maker for mature import competent peroxisomes). In the absence of Ceapin, ATF6α and PEX14 or Thiolase did not colocalize (*Figure 2A,C*, *Figure 2—figure supplement 2*). By contrast, in the presence of Ceapin, we observed ATF6α and PEX14 and Thiolase colocalization (*Figure 2A,C Figure 2—figure supplement 2*). Furthermore, in ABCD3 KD cells treated with Ceapin, ATF6α no longer formed foci or colocalized with peroxisomes (*Figure 2A,C*). This result was consistent in PEX19 KD cells, where peroxisome biogenesis is affected and ABCD3 is no longer chaperoned and targeted to the peroxisome (*Kashiwayama et al., 2007*; *Kashiwayama et al., 2005*; *Sacksteder et al., 2000*), ATF6α no longer formed foci in the presence

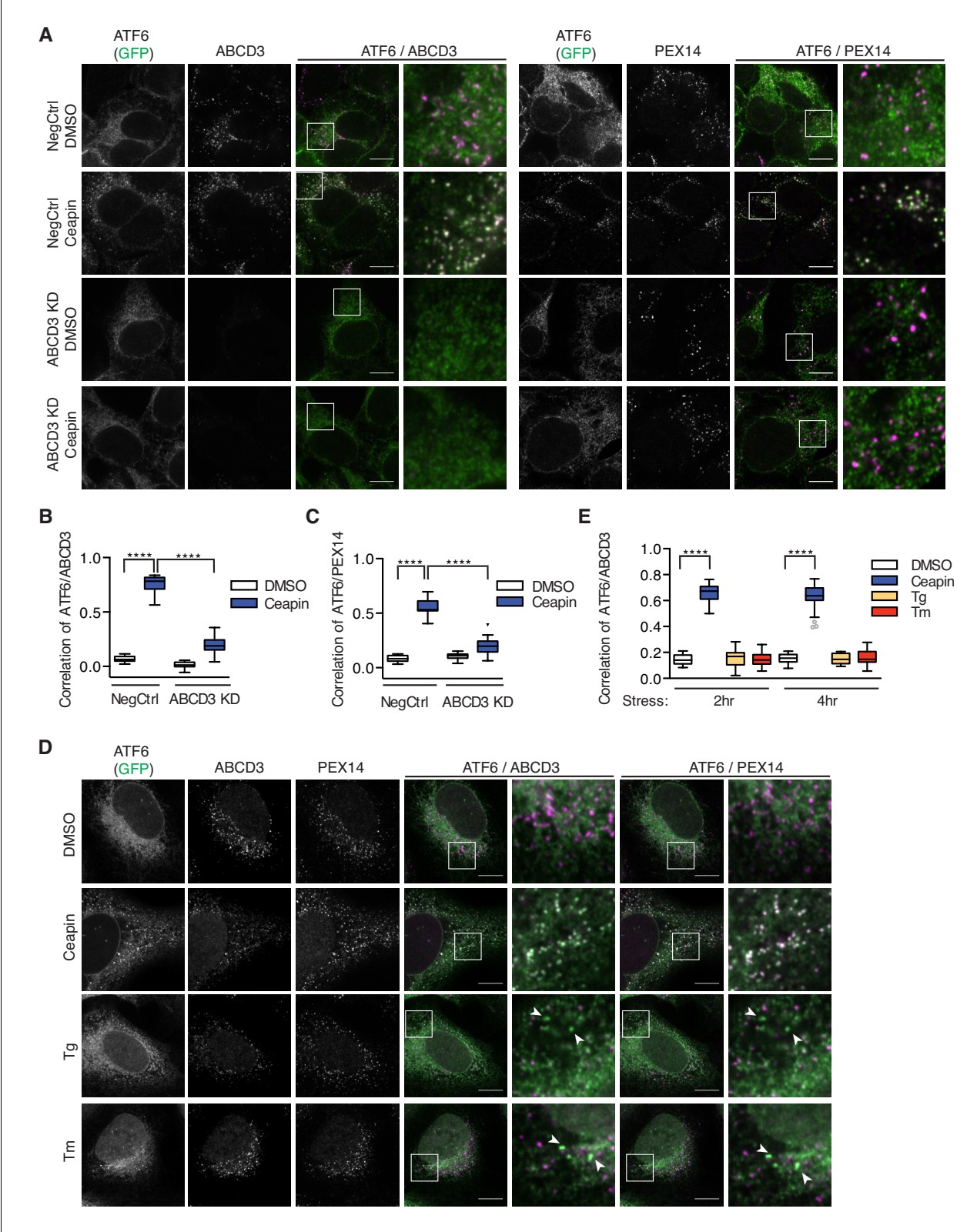

**Figure 2.** ABCD3 is required for Ceapin-induced ATF6α foci. (**A**) HEK293 CRISPRi cells stably expressing doxycycline inducible 3xFLAG-ATF6α with ABCD3 or NegCtrl KD were treated either with DMSO or Ceapin (6 μM) for 30 min prior to fixation, staining with anti-ABCD3 and/or anti-PEX14, and confocal fluorescent imaging. Scale bar, 10 μm. Images are representative of two independent experiments, in which we imaged at least 20 positions per well for each experiment. (**B and C**) Plotted is the mean and standard deviation of the mean per cell correlation of 3xFLAG-ATF6α and ABCD3 or

*Figure 2 continued on next page*

*Figure 2 continued*

PEX14 from (**A**) with at least 30 cells imaged per condition. All cells imaged in ABCD3 KD (96% KD), including wildtype cells, were used in quantification. Statistical analysis used unpaired two-tailed t-tests, **** indicates p<0.0001. (**D**) U2-OS cells stably expressing GFP-ATF6α were treated either with vehicle (DMSO), Tg (100 nM), Tm (2 µg/ml), or Ceapin (6 µM) for 2 hr or 4 hr (shown) prior to fixation, co-staining with anti-ABCD3 and anti-PEX14, and fluorescent imaging. Stress attenuated GFP-ATF6α foci are indicated by arrowheads. Scale bar, 10 µm. (**E**) Quantification of correlation of GFP-ATF6α and ABCD3 within PEX14 sites.

DOI: https://doi.org/10.7554/eLife.46595.006

The following figure supplements are available for figure 2:

**Figure supplement 1.** ABCD3 is not co-translationally translocated into the ER.

DOI: https://doi.org/10.7554/eLife.46595.007

**Figure supplement 2.** Ceapin-induced ATF6α foci colocalize with peroxisomal matrix protein Thiolase.

DOI: https://doi.org/10.7554/eLife.46595.008

**Figure supplement 3.** *PEX19* KD desensitizes cells to Ceapin and is required for Ceapin-induced ATF6α foci.

DOI: https://doi.org/10.7554/eLife.46595.009

of Ceapin (*Figure 2—figure supplement 3C*). Thus, peroxisomes interact with Ceapin-induced ATF6α foci in an ABCD3-dependent fashion to sequester ATF6α at the ER.

After prolonged ER stress, ATF6α attenuates and forms foci that are reminiscent of Ceapin induced foci (*Gallagher and Walter, 2016*). We next asked whether Ceapin was acting on the normal mechanism of ATF6α attenuation by testing ABCD3 colocalization with stress attenuated ATF6α foci. To induce stress attenuated ATF6α foci, we treated U2OS cells expressing GFP-ATF6α with ER stress, Tm or Tg (thapsigargin, which inhibits the ER calcium pump) for 2 and 4 hr. In positive control cells treated with Ceapin, ATF6α colocalized with ABCD3 and PEX14. In stress induced cells, attenuated ATF6α foci did not colocalize with ABCD3 or PEX14 by immunofluorescence (*Figure 2D–E*). Thus, Ceapin does not act on the ATF6α pathway by stabilizing the attenuated ATF6α state. The stress attenuated foci and Ceapin induced foci are distinct.

## Ceapin treatment does not inhibit ABCD3 activity

Since Ceapin treatment inhibits ATF6α, we next tested whether Ceapin treatment also inhibits ABCD3. ABCD3 knockout mice and hepatocytes display defects in bile acid biosynthesis (*Ferdinandusse et al., 2015*). To test if Ceapin treatment affects ABCD3 activity, we measured bile acid levels in a liver cancer cell line (HepG2) after Ceapin treatment and *ABCD3* KD. As expected, in ABCD3 KD cells, bile acid levels were decreased (*Figure 3*). In control cells treated at the $EC_{50}$ and ten-times the $EC_{50}$ of Ceapin, bile acid levels were similar to cells treated with vehicle only (*Figure 3*). Thus, Ceapin does not inhibit ABCD3 activity in cells.

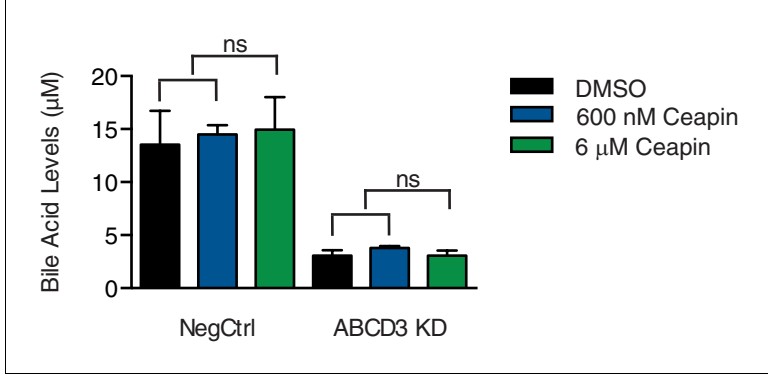

**Figure 3.** Ceapin treatment does not inhibit ABCD3 activity. Bile acid levels were measured in HepG2 CRISPRi cells with NegCtrl or ABCD3 KD treated with vehicle (DMSO), $EC_{50}$ of Ceapin (600 nM), and ten times the $EC_{50}$ of Ceapin-A7 (6 µM).

DOI: https://doi.org/10.7554/eLife.46595.010

## Ceapin-induced ATF6α-ABCD3 interaction does not require known ER-peroxisome tethers

The tight association between the ER and peroxisome is mediated by ER-peroxisome tethers, VAPA and VAPB on the ER and ACBD4 and ACBD5 on the peroxisomes (*Costello et al., 2017a*; *Costello et al., 2017b*; *Hua et al., 2017*). While the ER components are redundant, ACBD5 KD or overexpression alone decreases or increase ER-peroxisome contacts, respectively (*Costello et al., 2017a*; *Hua et al., 2017*). To determine whether proximity between the ER and peroxisomes induced by these tethers is required for Ceapin-induced foci formation, we knocked-down these known ER-peroxisome tethers. In tether KD cells treated with Ceapin, ATF6α foci still formed and ATF6α colocalized with ABCD3 (*Figure 4A–B*). Additionally, tether KD cells were not resistant to Ceapin treatment (*Figure 4C*), consistent with the results from our screen in which these components also did not score as hits.

## Ceapin-induced interactions do not require ER localized ATF6α nor ABCD3 transporter activity

We next tested if ER membrane association of ATF6α is required for Ceapin induced foci. To this end, we knocked down endogenous ATF6α and FACS sorted for a narrow, low level of GFP expression for truncated variants of ATF6α containing its cytosolic regions without the transmembrane and ER-lumenal domains (*Figure 5A*). We found that GFP-ATF6α(2-302), which was retained in the cytosol with a nuclear exit signal and was no longer associated with the ER, colocalized with ABCD3 and formed foci (*Figure 5A–B*). Further truncations showed that only the first 89 amino acids of ATF6α were both necessary and sufficient for Ceapin-dependent foci formation and colocalization with ABCD3 and peroxisomes (*Figure 5A–B*, *Figure 5—figure supplement 1*).

Since ABCD3 is a transporter, we then tested if ABCD3 catalytic activity was required for Ceapin action. Similarly to our ATF6α truncations, we also knocked down endogenous ABCD3 and FACS sorted for low level GFP expression of constructs with mutations of ABCD3 residues that mediate ATP binding (G478R) and hydrolysis (S572I) or a deletion of the entire catalytic domain (*Roerig et al., 2001*). There is one reported patient with a C terminal truncation of ABCD3 in which a reduced number of import competent peroxisomes are present (*Ferdinandusse et al., 2015*). Similarly, GFP-ABCD3ΔNBD cells, with a deletion of the entire catalytic domain, have reduced, enlarged peroxisomes (*Figure 5C*, *Figure 5—figure supplement 2*). We also confirmed correct localization of the GFP-ABCD3 constructs to the peroxisome (*Figure 5—figure supplement 2*). As a positive control, ABCD3 KD cells complemented with the full length ABCD3 construct were able to colocalize with and form ATF6α foci when treated with Ceapin (*Figure 5C–D*). In our catalytic activity mutants, we found that ABCD3 ATP binding or hydrolysis was not required for Ceapin-induced foci formation (*Figure 5C–D*). Although there are fewer larger peroxisomes in GFP-ABCD3ΔNBD cells, peroxisomal ABCD3 still induced foci formation and colocalized with ATF6α in the presence of Ceapin (*Figure 5C–D*). These results indicate that Ceapin-induced interactions do not require ER localized ATF6α nor ABCD3 transporter activity.

## Ceapin drives ATF6α-ABCD3 interaction in cells and in vitro

To identify components physically associating with ATF6α in the presence of Ceapin, we carried out native immunoprecipitation – mass spectrometric (IP-MS) analyses. We treated 3xFLAG-ATF6α HEK293 cells with Ceapin-A7 or an inactive analog, Ceapin-A5, in the presence of stress (Tg) and found that ABCD3 co-purified as the top hit with epitope-tagged ATF6α selectively in the presence of active Ceapin-A7 but not inactive Ceapin-A5 (*Figure 6A–B*). The native reciprocal affinity purification with full-length GFP-ABCD3 cells confirmed these results (*Figure 6C*). Furthermore, GFP-ABCD3ΔNBD, lacking the entire nucleotide binding domain, also physically associated with ATF6α in the presence of Ceapin (*Figure 6C*).

We then tested if the minimal cytosolic domain of ATF6α, GFP-ATF6α(2-90), physically associated with peroxisomal ABCD3. We immunoprecipitated GFP-ATF6α(2-90) from detergent solubilized lysates and specifically enriched ABCD3 in the presence of active Ceapin-A7 but not inactive Ceapin-A5 (*Figure 6D*). Thus, consistent with the above experiments where organelle tethering was not required, these results confirm that no other ER proteins are required for Ceapin-A7 induced ATF6α and ABCD3 physical association.

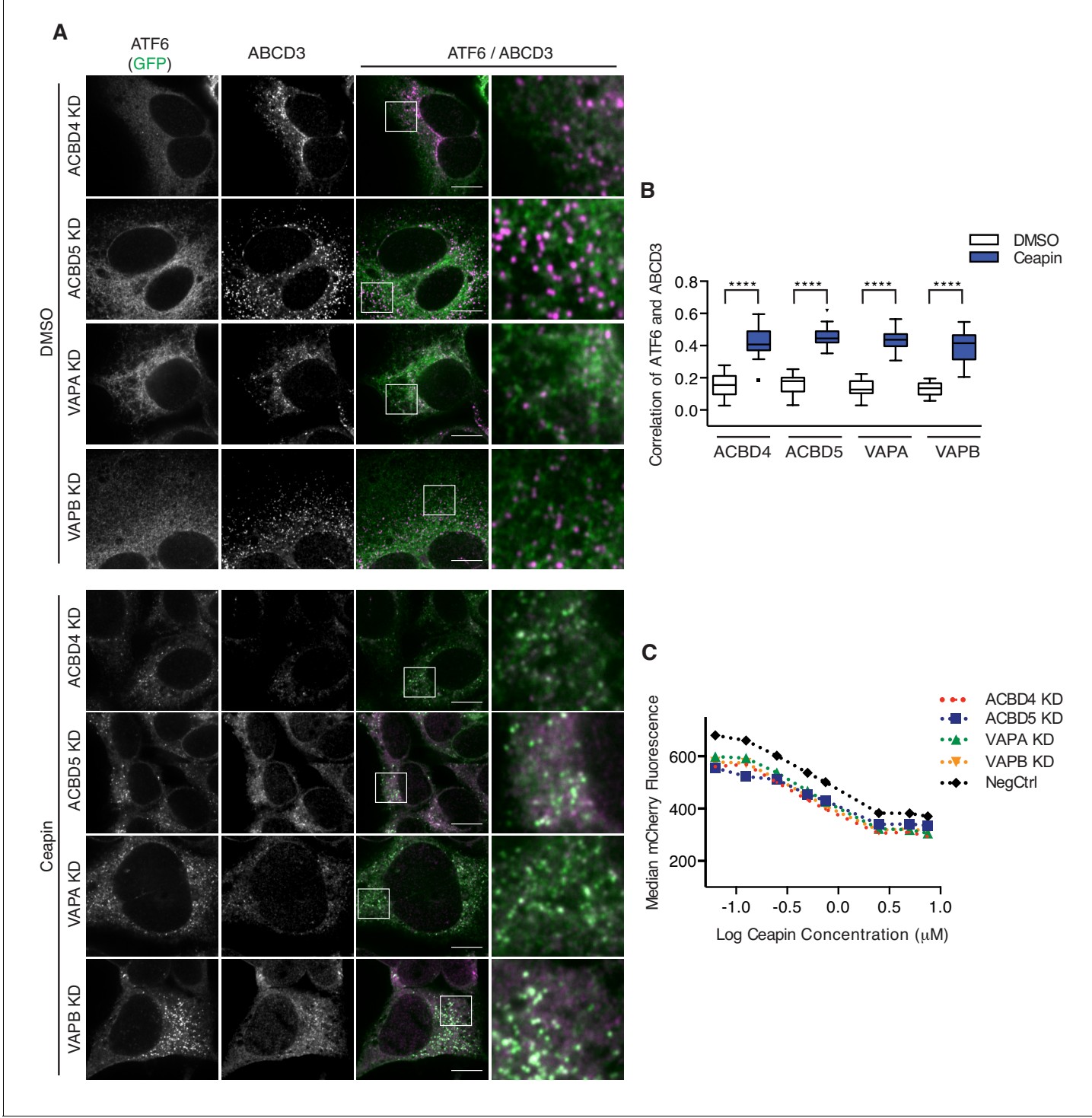

**Figure 4.** Ceapin-induced ATF6α-ABCD3 interaction does not require known ER-peroxisome tethers. (**A**) ER tether components, VAPA and VAPB, and peroxisome tether components, ACBD4 and ACBD5, were individually knocked-down in 3xFLAG-ATF6α HEK293 CRISPRi cell line, treated, fixed, and stained as in **Figure 2A** prior to fluorescence imaging. Scale bar, 10 μm. (**B**) Quantification of the correlation of ATF6α and ABCD3 from (**A**) and plotted as in **Figure 2B**. (**C**) K562 ERSE reporter cells with NegCtrl or indicated knockdowns were treated with Tm and increasing concentrations of Ceapin-A7 for 16 hr. Reporter fluorescence was measured by flow cytometry and median values were plotted (N = 3, ± SD).

DOI: https://doi.org/10.7554/eLife.46595.011

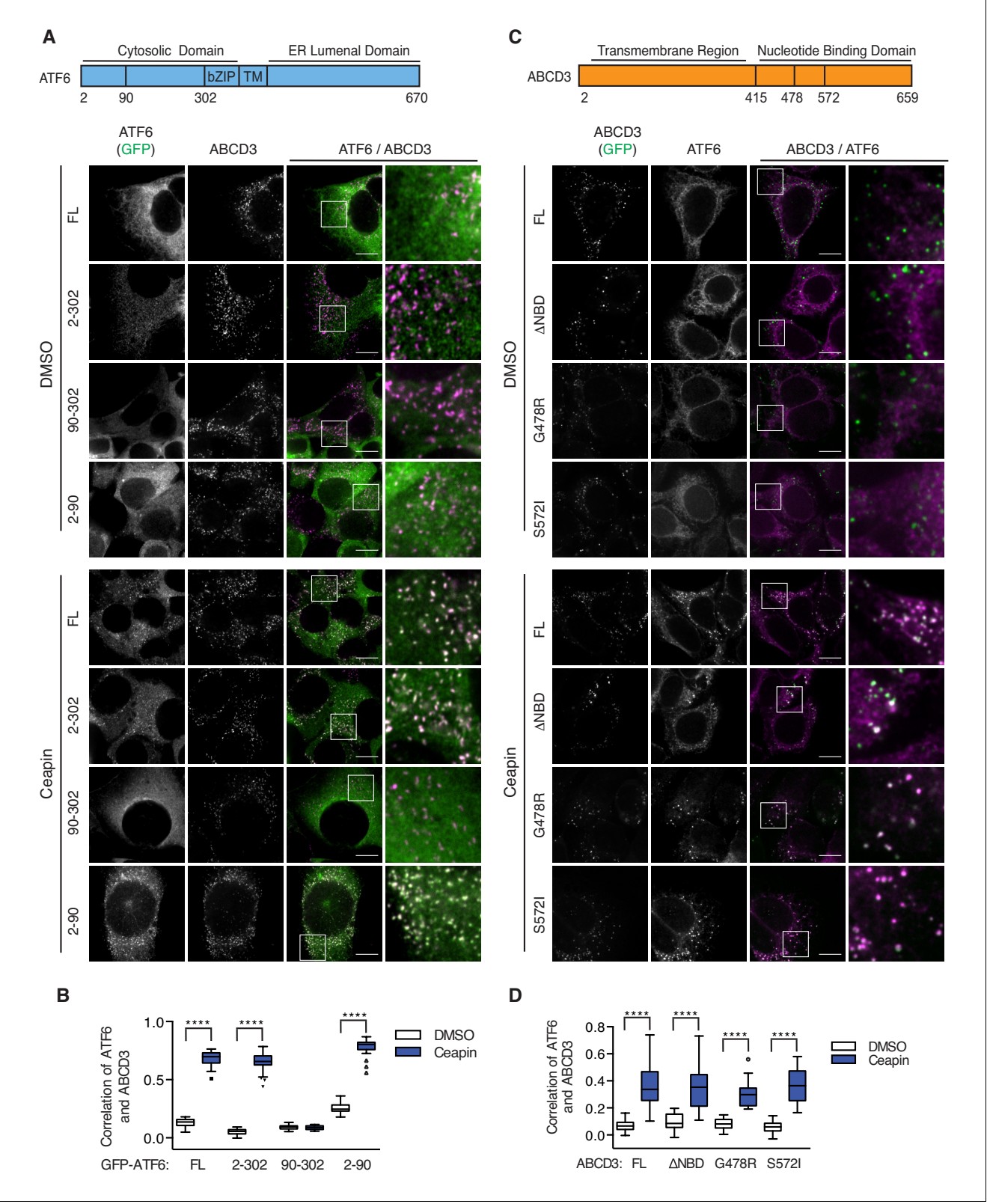

**Figure 5.** Ceapin-induced interactions do not require ER localized ATF6α nor ABCD3 transporter activity. (**A**) Diagram of GFP-ATF6α constructs tested. A nuclear exit signal (NES) was added to ATF6α truncated constructs to retain ATF6α in the cytosol. Endogenous ATF6α was knocked-down in 3xFLAG-ATF6α HEK293 CRISPRi cells grown without doxycycline, so that only GFP-ATF6α constructs were expressed. GFP-ATF6α-truncated cell lines were treated with DMSO or Ceapin-A7, fixed and stained for ABCD3. Scale bar, 10 μm. (**B**) Quantification of the correlation of GFP-ATF6α within

*Figure 5 continued on next page*

*Figure 5 continued*

ABCD3 sites. (C) Diagram of the GFP-ABCD3 mutants and truncations tested. Endogenous ABCD3 was knocked-down in 3xFLAG-ATF6α HEK293 CRISPRi cells so only GFP-ABCD3 constructs were expressed. GFP-ABCD3 cell lines were treated with DMSO or Ceapin-A7, fixed and stained for FLAG (ATF6α) (shown) and PEX14. Scale bar, 10 μm. (D) Quantification of the correlation of GFP-ABCD3 and 3xFLAG-ATF6α within PEX14 sites.

DOI: https://doi.org/10.7554/eLife.46595.012

The following figure supplements are available for figure 5:

**Figure supplement 1.** ATF6α(2-90) colocalizes with peroxisomal matrix protein Thiolase.

DOI: https://doi.org/10.7554/eLife.46595.013

**Figure supplement 2.** ABCD3 constructs localization to peroxisome.

DOI: https://doi.org/10.7554/eLife.46595.014

Finally, we tested whether purified ATF6α and ABCD3 were sufficient for Ceapin-induced tethering. In a binding assay with purified ATF6α(2-90) and ABCD3, our vehicle (DMSO) and inactive Ceapin-A5 controls did not induce ATF6α(2-90) and ABCD3 binding (*Figure 6E*). In the presence of Ceapin-A7, however, the cytosolic domain of ATF6α(2-90) and ABCD3 associated in solution (*Figure 6E*). Thus, Ceapin is directly responsible for tethering ABCD3 to ATF6α.

## Discussion

Ceapins, named for their ability to trap ATF6α in the ER, act with exquisite selectivity; they do not affect signaling of ATF6α's close homolog ATF6β or SREBP (sterol response element binding protein) (*Gallagher et al., 2016*), which depend on broadly used vesicular trafficking ER-Golgi pathways and are activated by the same Golgi-resident proteases (*Nadanaka et al., 2004*; *Okada et al., 2003*; *Schindler and Schekman, 2009*; *Ye et al., 2000*). Here we discovered the basis of this specificity. Ceapins induce neomorphic inter-organelle junctions, forcing interactions between the cytosolic domain of ER-tethered ATF6α and the peroxisomal transmembrane protein ABCD3 to sequester ATF6α from its normal trafficking route (*Figure 7*), and do so without interfering with or depending on ABCD3's normal function. Since ABCD3 protein expression is ten-fold higher than ATF6 (*Hein et al., 2015*), it is likely ABCD3 is not saturated. Ceapin induced interaction of ABCD3 with the most N-terminal region of ATF6α also clarifies how ATF6α foci are excluded from COPII trafficking, while the transmembrane region of ATF6α remains accessible to protease cleavage. Mechanistically, Ceapins could act as molecular staples that physically bridge the respective proteins or bind to one or the other inducing allosteric changes that promote their association; but in either case, Ceapin is responsible for tethering ABCD3 to ATF6α.

Remarkably, in the absence of Ceapins, ATF6α and ABCD3 localize to different parts of the cell and are not known to interact physically or functionally. Indeed, an 89-amino acid fragment of ATF6α fused to GFP is sufficient to recruit GFP to peroxisomes, ruling out the need for endogenous inter-organellar tethers. This Ceapin-induced tethering enables an 'anchor away' strategy but one that uses an abundant, ubiquitously expressed endogenous acceptor protein. There has been increasing interest in small molecules that induce novel protein-protein interactions with therapeutic potential (*de Waal et al., 2016*; *Han et al., 2017*; *Krönke et al., 2014*; *Lu et al., 2014*; *Krönke et al., 2015*; *Petzold et al., 2016*; *Uehara et al., 2017*). Ceapins provide a novel example of such molecules and increase the repertoire to include the induction of inter-organellar connections, opening new mechanistic routes for drug development and synthetic biology by broadly enabling control of protein function through chemical-induced misdirection.

Understanding the mechanism of action of a chemical modulator of cellular stress and establishing that it is acting directly and specifically is critical for exploiting the utility of any stress modulators either as research or potential therapeutic agents. Our identification of the mechanism by which Ceapins achieve their remarkable specificity forms a foundation to explore the utility of ATF6α inhibition in the treatment of cancers, such as squamous carcinomas, in which ATF6α signaling protects dormant tumor cells from classical chemotherapies (*Schewe and Aguirre-Ghiso, 2008*).

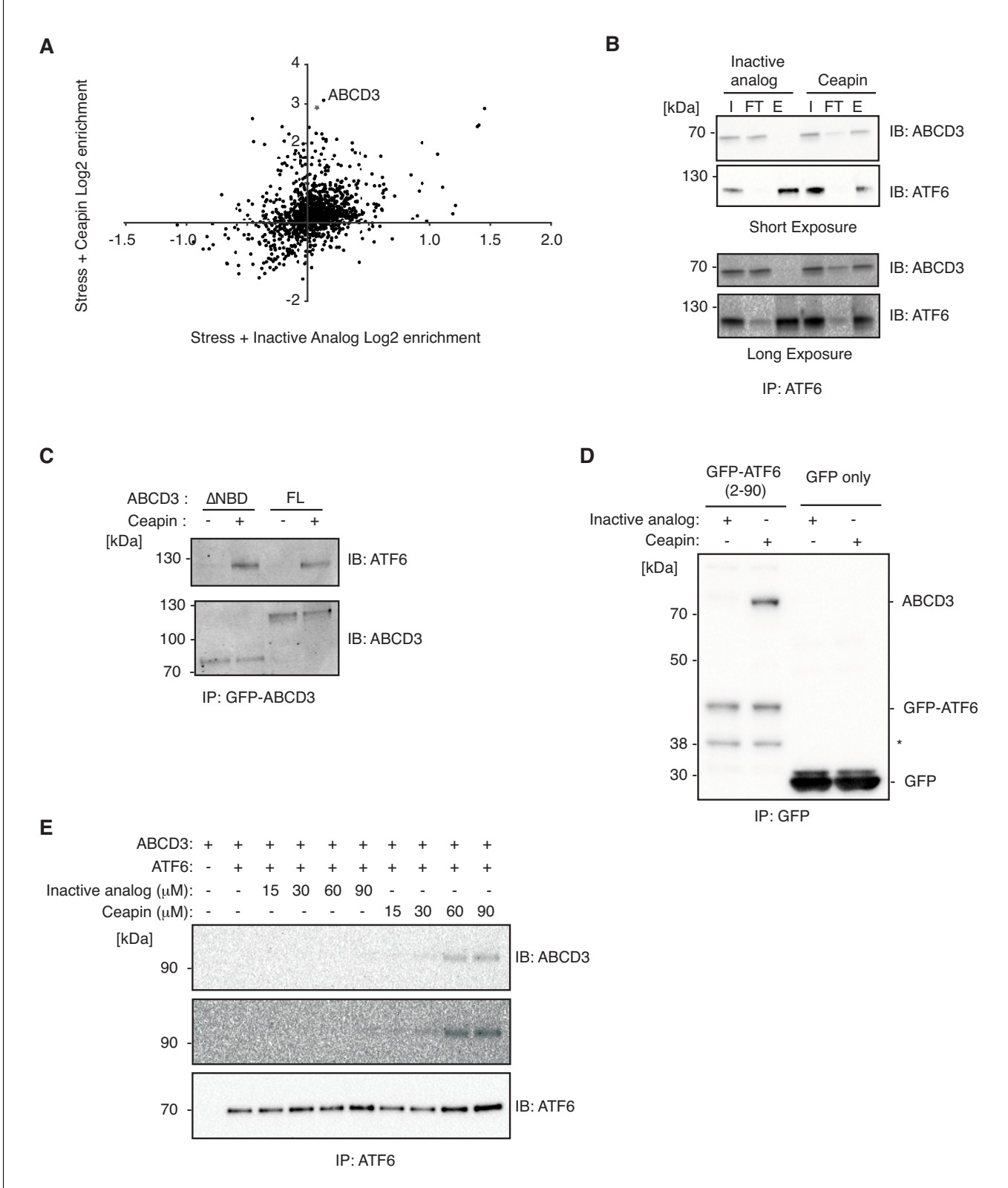

**Figure 6.** Ceapin drives ATF6α-ABCD3 interaction in cells and in vitro. (**A and B**) Proteomic analysis and immunoblot (IB) of anti-FLAG affinity purification from 3xFLAG-ATF6α HEK293 cells treated with stress (100 nM Tg) and inactive Ceapin-A5 analog (6 μM) or active Ceapin-A7 (6 μM) with two replicates for each treatment condition. The proteins identified with affinity-purified FLAG-ATF6 treated with ER stress and Ceapin-A5 or Ceapin-A7 are listed in *Figure 6—source data 1*. SQSTM1 KD (*) was the top second hit in proteomics, however, SQSTM1 KD in the K562 ATF6 reporter cell

*Figure 6 continued on next page*

*Figure 6 continued*

line did not render cells resistant to Ceapin treatment and retained a similar response to negative control cells. I, input; FT, flow-through; E, elution. (C) Immunoprecipitation of full-length GFP-ABCD3 and GFP-ABCD3ΔNBD from cells treated with DMSO or Ceapin-A7. (D) Detergent solubilized GFP-ATF6α(2-90) or GFP-only cell lysates were incubated with Ceapin-A7 or inactive analog Ceapin-A5 and affinity purified with anti-GFP. (*) Indicates a degradation product. (E) Purified ATF6α-MBP and ABCD3-GFP were incubated with inactive Ceapin-A5 or active Ceapin-A7 and affinity purified with anti-MBP antibody.

DOI: https://doi.org/10.7554/eLife.46595.015

The following source data is available for figure 6:

**Source data 1.** Excel spreadsheet showing all the proteins identified with affinity-purified FLAG-ATF6 treated with ER stress and Ceapin-A5 or Ceapin-A7.
DOI: https://doi.org/10.7554/eLife.46595.016

## Materials and methods

### Cell culture and experimental reagents

U2OS, 293TREx, and HepG2 cells were cultured in DMEM. K562 cells were cultured in RPMI. Culture media was supplemented with 10% fetal bovine serum (FBS), 1% L-glutamine, and 1% penicillin/streptomycin (ThermoFisher). U2-OS cells stably expressing GFP-HsATF6α were purchased from Thermo Scientific (084_01) and supplemented with 500 μg/ml G418 to maintain GFP-HsATF6α expression. HeLa CRISPRi cells expressing SFFV-dCas9-BFP-KRAB were previously described (*Jost et al., 2017*). Tunicamycin and thapsigargin were purchased from Sigma. Antibodies used were rabbit anti-GFP (ThermoFisher A11122), mouse anti-FLAG M2 (Sigma F1804), rat anti-GRP94 9G10 (abcam ab2791), rabbit anti-ACAA1 (Sigma HPA007244), rabbit anti-pmp70 (ab109448) for PFA fixation and (PA1-650) for methanol fixation, mouse anti-pmp70 (ab211533) for PFA fixation and (SAB4200181) for methanol fixation.

### Generation of constructs and cell lines

To generate CRISPRi knockdown cell lines, SFFV-dCas9-BFP-KRAB (Addgene 46911) or UCOE-EF1α-dCas9-BFP-KRAB (*Jost et al., 2017*) were stably transduced and FACS-sorted for BFP positive cells. 293 TREx cells expressing doxycycline-inducible 6xHis-3xFLAG-HsATF6α (*Gallagher et al.,*

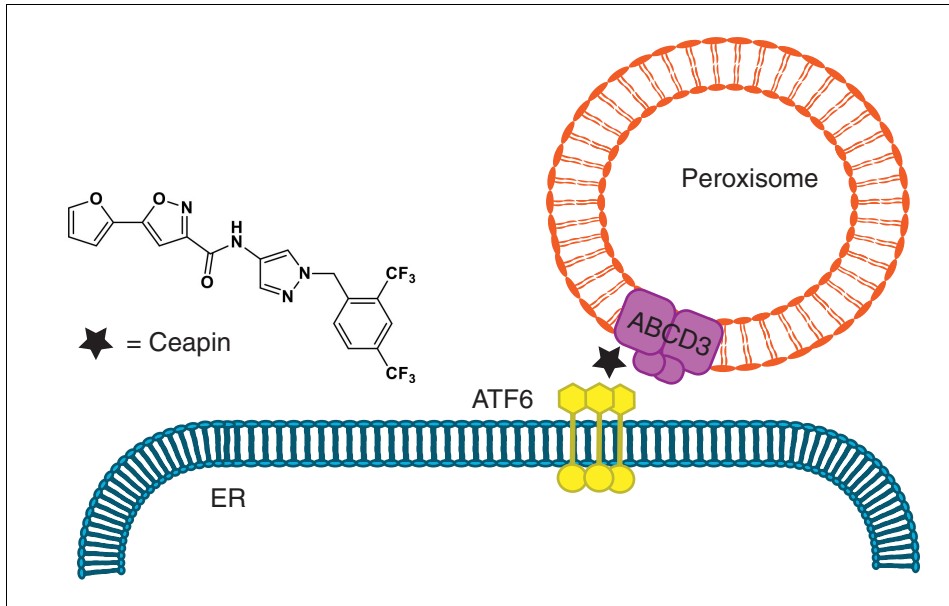

**Figure 7.** Model for Ceapin induced ATF6α inhibition. Ceapins sequester ATF6α into a transport-incompetent pool during ER stress by tethering ATF6α to peroxisomal ABCD3. ATF6α is occluded from COPII trafficking, while its transmembrane domain remains accessibly to protease cleavage.
DOI: https://doi.org/10.7554/eLife.46595.017

*2016*) were infected with SFFV-dCas9-BFP-KRAB and sorted twice for BFP expressing cells. HepG2 cells from ATTC (CRL-10741) were infected with UCOE-EF1α-dCas9-BFP-KRAB and FACS sorted for BFP expression. The ERSE reporter construct was generated by subcloning mCherry into the ERSE. Fluc.pcDNA3.1 (*Mortenson et al., 2018*) using the polymerase incomplete primer extension method to replace the FLuc gene. This construct was then subcloned into ClaI and EcoRI digested pLenti6. V5.GFP. K562 cells stably expressing dCas9-KRAB (*Gilbert et al., 2014*) were stably transduced with the ERSE reporter construct and a monoclonal line was selected and expanded to generate K562 ERSE reporter cell line.

Individual gene knockdowns were carried out by selecting sgRNA protospacers from compact hCRISPRi-v2 library and cloning into lentiviral plasmid pU6-sgRNA EF1α-puro-t2a-BFP (Addgene 60955) as previously described (*Horlbeck et al., 2016*). Protospacer sequences used for individual knockdowns are listed in *Table 1*. The resulting sgRNA expression vectors were packaged into lentivirus by transfecting HEK293T with standard packaging vectors using TransIT-LTI Transfection Reagent (Mirus, MIR 2306). The viral supernatant was harvested 2–3 days after transfection and frozen prior to transduction into CRISPRi knockdown cell lines described above.

3xFLAG-ATF6α HEK293 CRISPRi described above was stably transduced with sgRNA knockdown of endogenous ATF6α or ABCD3 and grown without doxycycline to ascertain that only truncation constructs would be expressed. ATF6α truncation constructs were generated by Gibson assembly of IDT gblock of sfGFP and ATF6α PCR amplified from peGFP-HsATF6α (Addgene #32955) into inserted into BamHI/NotI digested pHR-SFFV-Tet3G (*Gilbert et al., 2014*). ATF6α truncations were PCR amplified with reverse primers containing nuclear exit signal (NES) (NES, CTGCCCCCCC TGGAGCGCCTGACCCTG; NES_REV, CCCCTGCAGCTGCCCCCCCTGGAGCGGCTGACCCTG) to retain ATF6α in the cytosol. Full length GFP-ABCD3 and GFP-ABCD3ΔNBD (2-416) were cloned by Gibson assembly of ABCD3 PCR amplified from cDNA and IDT gblock of sfGFP into BamHI/NotI digested pHR-SFFV-Tet3G (*Gilbert et al., 2014*). ABCD3 G478R and S572I mutations were generated by site directed mutagenesis (QuikChange Lightning Agilent) of full-length GFP-ABCD3 construct. ATF6α and ABCD3 truncation vectors were packaged into lentivirus as described above, stably transduced, and FACS sorted for a narrow and low level of GFP expression.

U2OS Flp-In cells were infected with UCOE-EF1α-dCas9-BFP-KRAB and FACS sorted for BFP expression. They were then stably transduced with sgRNA knockdown of endogenous ATF6α and GFP-ATF6α(2-90) construct, and FACS sorted for a narrow level of GFP expression. Parental cell lines and commercially available cell lines were authenticated by STR analysis and tested negative for mycoplasma contamination.

## Genome-scale CRISPRi screen

Reporter screens were carried out using protocols similar to those previously described (*Adamson et al., 2016*; *Gilbert et al., 2014*; *Sidrauski et al., 2015*). The compact (five sgRNA/ gene) hCRISPRi-v2 (*Horlbeck et al., 2016*) sgRNA libraries were transduced into ERSE reporter cells

**Table 1.** Protospacer sequence of sgRNAs.

| Gene | Protospacer |
| --- | --- |
| NegCtrl | GCGCCAAACGTGCCCTGACGG |
| ATF6 | GTGGGATCTGAGAATGTACCA |
| ABCD3-1 | GGTACCAGCGAGCCGGCGAG |
| ABCD3-2 | GACTGCCGGTACCAGCGAGC |
| PEX19-1 | GGCCGAAGCGGACAGGGAAT |
| PEX19-2 | GGAGGAAGGCTGTAGTGTCG |
| ACBD4 | GCCGGCCCTGCTGGACCCCG |
| ACBD5 | GGGAGCCGCTCTCCCACCCT |
| VAPA | GCACCGAACCGGTGACACAG |
| VAPB | GCGGGGGTCCTCTACCGGGT |

DOI: https://doi.org/10.7554/eLife.46595.018

at a MOI <1 (55% BFP+ cells). Cells were grown in spinner flasks for 2 days without selection, selected with 2 µg/ml puromycin for 2 days, and allowed to recover for 3 days. Cells were then split into two populations, which were treated for 16 hr with 6 µg/ml tunicamycin alone or 6 µg/ml tunicamycin and 3 µM Ceapin (EC90). Cells were then sorted based on reporter fluorescence using BD FACS Aria2. Cells with the highest (~30%) and lowest (~30%) mCherry expression were collected and frozen after collection. Approximately 20 million cells were collected per bin. Genomic DNA was isolated from frozen cells, and the sgRNA-encoded regions were enriched, amplified, and prepared for sequencing.

Sequenced protospacer sequences were aligned and data were processed as described (*Gilbert et al., 2014*; *Horlbeck et al., 2016*) with custom Python scripts (available at https://github.com/mhorlbeck/ScreenProcessing). Reporter phenotypes for library sgRNAs were calculated as the log2 enrichment of sgRNA sequences identified within the high-expressing mCherry over the low-expressing mCherry cells. Phenotypes for each transcription start site were then calculated as the average reporter phenotype of all five sgRNAs. Mann-Whitney test p-values were calculated by comparing all sgRNAs targeting a given TSS to the full set of negative control sgRNAs. For data presented in *Figure 1B*, screen hits are defined as those genes where the absolute value of a calculated reporter phenotype over the standard deviation of all evaluated phenotypes multiplied by the log10 of the Mann-Whitney p-value for given candidate is greater than 7. Growth screen data (*Horlbeck et al., 2016*) was used to label genes with growth phenotype of at least −0.19. Ceapin independent genes are defined as genes that were hits in tunicamycin alone and tunicamycin with Ceapin treatment since their phenotype was independent of Ceapin treatment. Genes involved in chromatin remodeling and architecture have been previously described in UPR screens to act downstream and directly affect expression of the reporter (*Jonikas et al., 2009*). Chromatin related genes that impact reporter expression are labeled with (★) in *Figure 1—figure supplement 1C–D*.

## Bile acid assay

HepG2 CRISPRi ABCD3 KD and NegCtrl cells were treated with DMSO or Ceapin at 600 nM or 6 µM for 24 hr. Cells were harvested in scrapping buffer (cold PBS with 10 µM MG132 and 1X protease inhibitor), spun down, resuspended in lysis buffer (50 mM Tris pH 7.4, 150 mM NaCl, 5 mM EDTA, 1X protease inhibitor, and 1% LMNG), and spun down at 10,000 x g for 10 min. The supernatant was used for bile acid assay (Cell Biolabs STA-631) as described by the manufacturer.

## Quantitative RT-PCR

Cells were harvested and total RNA was isolated using the NucleoSpin RNA II (Macherey-Nagel) according to manufacturer's instructions. RNA was converted to cDNA using AMV reverse transcriptase under standard conditions with oligo dT and RNasin (Promega, Life Technologies). Quantitative PCR reactions were prepared with a 2x master mix according to the manufacturer's instructions (KAPA SYBR FAST qPCR Kit). Reactions were performed on a LightCycler thermal cycler (Roche). Primers used were against HSPA5 (forward, TGTGCAGCAGGACATCAAGT: reverse, AGTTCCAGCGTCTTTGGTTG) and HSP90B1 (forward, GGCCAGTTTGGTGTCGGTTT; reverse, CGTTCCCCGTCCTAGAGTGTT).

## Immunofluorescence

Fluorescence confocal imaging was carried out as described in *Gallagher and Walter (2016)*. 293 TREx, U2OS, HepG2, and HeLa cells were plated in 8-well ibiTreat µSlide (ibidi 80826) at 20–25,000 cells/well. In 3xFLAG-ATF6α imaging experiments (*Figure 2A–C*, *Figure 2—figure supplements 2–3*, *Figure 4A–C*, *Figure 5A–D*, *Figure 5—figure supplement 2*), 3xFLAG-ATF6α HEK293 CRISPRi cells were plated and induced with 50 nM doxycycline on the same day. On the following day, cells were treated with DMSO or 6 µM Ceapin for 30 min and then fixed with cold methanol or 4% PFA. For cold methanol fixation, media was removed, cold ethanol was added for 3 min at −20 °C, washed, and permeabilized with PHEM (60 mM PIPES, 25 mM HEPES, 10 mM EGTA, 2 mM MgCl₂, pH 6.9) with 0.1% Triton X-100, and washed twice with PHEM. For PFA fixation, media was removed from slides, 4% PFA (EMS) was added for 10 min at room temperature, washed, permeabilized as above, and washed with PHEM. Slides were then treated with blocking buffer (5% goat serum (Jackson ImmunoResearch) in PHEM) for 1 hr at room temperature. Antibodies were diluted in blocking

buffer and incubated with cells at 4°C overnight. After three washes with PHEM, cells were incubated with secondary antibodies conjugated to Alexa 488, Alexa 568, and/or Alexa 633 (Invitrogen) for 1 hr at room temperature. Slides were imaged on a spinning disk confocal with Yokogawa CSUX A1 scan head, Andor iXon EMCCD camera and 100x ApoTIRF objective NA 1.49 (Nikon). Linear adjustments were made using ImageJ. Quantification of correlation between ATF6α with ABCD3, Thiolase, and/or PEX14 was calculated using CellProfiler 2.1.1. ABCD3, Thiolase, or PEX14 images were used to identify objects, a background threshold for ATF6α images was set to 1.2, and clumped objects were separated based on intensity. The resulting ABCD3, Thiolase, or PEX14 outlines were used as masks to count the ATF6α intensity within ABCD3, Thiolase, or PEX14. Data from CellProfiler was imported into GraphPad Prism version 6.0 for statistical analysis and plotting.

## Nuclear translocation assay

3xFLAG-ATF6α HEK293 CRISPRi cells with ABCD3 KD and ABCD3 KD complemented with full length GFP-ABCD3 construct were plated in ibidi 96-well ibiTreat μ-plate (ibidi 89626) and induced with 50 nM doxycycline on the same day. On the following day, cells were treated with DMSO or 100 nM Tg for 2 hr and then fixed with 4% PFA as described above. The plates were then treated with blocking buffer (5% goat serum (Jackson ImmunoResearch) in PHEM) for 1 hr at room temperature. Primary antibodies, mouse anti-FLAG M2 (Sigma F1804) and rat anti-GRP94 9G10 (abcam ab2791), were diluted in blocking buffer and incubated with cells at 4°C overnight. After three washes with PHEM, cells were incubated with secondary antibodies conjugated to Alexa 568 and Alexa 633 (Invitrogen) and nuclear stain (DAPI, Molecular Probes D-1306, 5 μg/mL) for 1 hr at room temperature. Quantification ATF6α signal in ER and nucleus was calculated using CellProfiler 2.1.1 as described in *Gallagher and Walter (2016)*. DAPI images were used to identify primary objects and clumped objects were distinguished based on fluorescence intensity. The GRP94 images were then used to generate secondary objects from primary objects using global Otsu two-class thresholding with weighted variance. The final ER mask was generated by subtracting the nuclear area from the ER area. Lastly, the FLAG-ATF6α images were used to calculate FLAG-ATF6α intensity in the nucleus and ER and determine the nucleus to ER ratio of each cell. Data from CellProfiler was exported as a MATLAB file for analysis and plotted on GraphPad Prism version 6.0.

## Immunoprecipitation and immunoblot analysis

Cells were grown in 100 mm plates with two replicates for each treatment condition, treated with 50 nM doxycycline the following day, treated with 100 nM Tg and 6 μM Ceapin-A5 or 6 μM Ceapin-A7 for 30 min on the day of harvest, and harvested in scrapping buffer (cold PBS with 10 μM MG132 and 1X protease inhibitor). Ceapin A-7, inactive analog Ceapin A-5, or DMSO were kept in scrapping and lysis buffers throughout IP. Cells were lysed for 1 hr at 4°C in lysis buffer (50 mM Tris pH 7.4, 150 mM NaCl, 5 mM EDTA, 1X protease inhibitor, and 1% LMNG (Anatrace NG322)). The lysates were cleared by centrifugation at 17,000 x g for 30 min. Dynabeads Protein-G (ThermoFisher) were bound with Sigma FLAG M2 antibody for 1 hr at 4°C and crosslinked with 100 μM BS3 crosslinker for 30 min. 293 TREx 3XFLAG cell lysates were then incubated with these FLAG beads for 2 hr at 4°C. IP beads were washed with wash buffer (lysis buffer without LMNG) and boiled and eluted in buffer containing 50 mM Tris pH 6.8, 300 mM NaCl, 2% SDS, and 10 mM EDTA. Protein samples were then precipitated, trypsin digested, labeled with tandem mass tags (TMT), and analyzed by liquid chromatography-mass spectrometry using Multidimensional Protein Identification Technology (MuD-PIT), as described previously (*Mortenson et al., 2018*; *Plate et al., 2018*). TMT intensities for proteins detected in each channel were normalized to the respective TMT intensity of ATF6α. TMT ratios for individual proteins were then calculated between Tg+Ceapin-A7/DMSO treatment or Tg+Ceapin-A5/DMSO treatment.

The reciprocal affinity purification with full-length GFP-ABCD3 or GFP-ABCD3ΔNBD cells was carried out by culturing, treating, and lysing cells as described above. 293 TREx 3XFLAG GFP-ABCD3 clarified cell lysate was then incubated with GFP-Trap_MA ChromoTek beads for 2 hr at 4°C. IP beads were washed with wash buffer (lysis buffer without LMNG) and boiled in SDS sample buffer for 10 min.

Cells for in vitro incubation were lysed with lysis buffer containing LMNG (described above) and cleared by centrifugation at 17,000 x g for 30 min in the absence of any drug. Cleared supernatant

was then incubated with Ceapin A-7 or inactive analog Ceapin A-5 for 30 min at room temperature, bound to GFP-Trap_MA ChromoTek beads for 1 hr at 4℃, washed with wash buffer containing Ceapin A-7 or Ceapin A-5, and eluted by boiling in SDS sample buffer.

For in vitro binding studies with purified components, 6.25 nM 3XFLAG-ATF6α (2-90)-TEV-MBP-HIS$_{6X}$ and 100 nM ABCD3-eGFP- HIS$_{8X}$ were incubated in lysis buffer (50 mM Tris pH 7.4, 150 mM NaCl, 5 mM EDTA, 1X protease inhibitor, 0.001% LMNG) with 15–90 µM Ceapin-A7 or inactive Ceapin-A5 for 30 min at room temperature. Samples were then incubated with MBP-Trap_A ChromoTek beads for 1 hr at 4℃, washed with same buffer containing Ceapin A-7 or A-5 and 300 mM NaCl, and eluted by boiling in SDS sample buffer.

Samples were run on a precast 4–12% Bis-Tris polyacrylamide gel (Life Technologies) under denaturing conditions and transferred to nitrocellulose membrane. Antibodies described above for FLAG, GFP, and Pmp70 (SAB4200181) were used to detect proteins and blots were imaged for chemiluminescence detection using a ChemiDocTM XRS + Imaging System (Bio-Rad) (*Figure 6B and D–E*) or LICOR system (*Figure 6C*).

## Generation of recombinant proteins

Human ATF6α(2-90) with an N-terminal 3XFLAG was cloned into pET16b-TEV-MBP-HIS$_{6X}$ (Novagen) using Gibson assembly. The construct was expressed in in BL21-Gold(DE3) *E. coli* cells, grown to 0.6–0.8 OD$_{600}$, and induced overnight at 16℃ with 0.25 mM IPTG (Gold Biotechnology). The cells were harvested and resuspended in buffer containing 50 mM HEPES pH 7, 150 mM NaCl, 10% glycerol, 2 mM TCEP, and complete EDTA-free protease inhibitor cocktail (Roche). After lysis by sonication, the lysate was clarified at 30,000 x g for 30 min at 4℃. The clarified lysate was loaded onto a HisTrap HP 5 ml column, washed in binding buffer (50 mM HEPES, pH 7, 300 mM NaCl, 1 mM TCEP, 10% glycerol, and 25 mM imidazole), and eluted with a linear gradient of 25 mM to 1M imidazole in the same buffer. The ATF6α fractions eluted at 240 mM imidazole were collected and concentrated with an Amicon Ultra-15 concentrator (EMD Millipore) with a 30,000-dalton molecular weight cutoff. The ATF6α concentrated fraction was loaded onto a Mono Q HR16/10 column (GE Healthcare), washed in Buffer A (50 mM HEPES, pH 7.5, 100 mM NaCl, 10% glycerol, and 1 mM DTT) and eluted with a linear gradient of 100 mM to 1M NaCl in the same buffer. Fractions were collected, concentrated as above, and loaded onto a Superdex 200 10/300 GL column (GE Healthcare) equilibrated with buffer containing 30 mM HEPES, pH 7.5, 300 mM NaCl, 5% glycerol, and 1 mM DTT.

Expression and purification of human ABCD3: Full-length human ABCD3 isoform I was synthesized and cloned into modified pFastBac1 plasmid with a C-terminal -eGFP −8XHis-tag for baculoviral expression in *Spodoptera frugiperda* SF9 cells. Bacmid DNA was produced by transforming the recombinant pFastBac1 plasmid into *E. coli* DH10Bac strain. To express the protein, SF9 cells were infected with the bacmid made from recombinant pFastBac1 plasmid at multiplicity of infection (MOI) = 2 for 48 hr at 27℃. The cells were harvested and resuspended in lysis buffer (50 mM Tris Cl, pH 7.5, 100 mM NaCl, 100 mM MgCl$_2$, 10% glycerol) containing complete EDTA-free protease inhibitor cocktail (Roche), and lysed by sonication. The lysate was centrifuged at 186,010 x *g* for 2 hr to extract the membrane fraction. 3 g of the membrane was solubilized in 30 ml of lysis buffer containing 1% w/v lauryl maltose neopentyl glycol (LMNG) (Anatrace): 0.1% w/v cholesteryl hemisuccinate (CHS) (Anatrace) overnight. Solubilized membrane was clarified by centrifugation at 104,630 x *g* for 30 min with 5 mM imidazole added. A HiTrap TALON crude 1 ml column (GE Healthcare) was equilibrated with the lysis buffer containing 5 mM imidazole and solubilized membrane loaded onto the column. After binding the column was washed with 15 ml of 10 mM imidazole, 0.02% glyco-diosgenin (GDN) (Anatrace) in lysis buffer. The protein was eluted from the column with 10 ml of 150 mM imidazole, 0.02% GDN containing lysis buffer. The protein obtained was concentrated using Amicon Ultra-15 centrifugal filter units (MilliporeSigma) and size exclusion chromatography was done to further purify the protein in SEC buffer (20 mM HEPES, pH 7.5, 100 mM NaCl, 2 mM MgCl$_2$, 2% glycerol and 0.02% GDN).

## Acknowledgements

We thank Marco Jost for critical reading of the manuscript; Jeff Kelly and Luke Wiseman for advice on target identification and mass spectrometry; Mable Lam, James Nuñez, and Elif Karagöz for

advice on protein purification; Nico Stuurman and Vladislav Belyy for advice on fluorescence microscopy; John Christianson for advice on solubilization of intact membrane protein complexes; and members of the Walter and Weissman labs for helpful discussions. This research was supported by Collaborative Innovation Awards from the Howard Hughes Medical Institute (HHMI) and by NIH/NIGMS New Innovator Award DP2 OD021007 (MK). Research of MG and RMS is supported by NIH GM111126 to RMS PW and JSW are Investigators of the HHMI.

## Additional information

### Funding

| Funder | Grant reference number | Author |
|--------|------------------------|--------|
| Howard Hughes Medical Institute | | Jonathan S Weissman Peter Walter |
| National Institutes of Health | GM111126 | Robert M Stroud |
| National Institutes of Health | DP2 OD021007 | Martin Kampmann |

The funders had no role in study design, data collection and interpretation, or the decision to submit the work for publication.

### Author contributions

Sandra Elizabeth Torres, Conceptualization, Data curation, Formal analysis, Validation, Investigation, Methodology, Writing—original draft; Ciara M Gallagher, Lars Plate, Formal analysis, Investigation, Methodology, Writing—review and editing; Meghna Gupta, Christina R Liem, Xiaoyan Guo, Ruilin Tian, Investigation, Methodology, Writing—review and editing; Robert M Stroud, Martin Kampmann, Resources, Supervision, Writing—review and editing; Jonathan S Weissman, Conceptualization, Supervision, Funding acquisition, Writing—review and editing; Peter Walter, Conceptualization, Resources, Supervision, Funding acquisition, Writing—review and editing

### Author ORCIDs

Sandra Elizabeth Torres (iD) https://orcid.org/0000-0003-2764-4407
Martin Kampmann (iD) http://orcid.org/0000-0002-3819-7019
Jonathan S Weissman (iD) https://orcid.org/0000-0003-2445-670X
Peter Walter (iD) https://orcid.org/0000-0002-6849-708X

### Decision letter and Author response

Decision letter https://doi.org/10.7554/eLife.46595.021
Author response https://doi.org/10.7554/eLife.46595.022

## Additional files

### Supplementary files

• Transparent reporting form
DOI: https://doi.org/10.7554/eLife.46595.019

### Data availability

All data generated or analyzed during this study are included in the manuscript and supporting files.

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
