## [Decision Letter]

Thank you for submitting your article "Ceapins block the unfolded protein response sensor ATF6α by inducing a neomorphic inter-organelle tether" for consideration by *eLife*. Your article has been reviewed by three peer reviewers, including Elizabeth A Miller as the Reviewing Editor and Reviewer #1, and the evaluation has been overseen by Vivek Malhotra as the Senior Editor.

The reviewers have discussed the reviews with one another and the Reviewing Editor has drafted this decision to help you prepare a revised submission.

Summary:

This manuscript from the Walter and Weissman labs aims to understand the mechanism of action of a UPR modulator, Ceapin, which acts on ATF6 to prevent its transport to the Golgi and subsequent activation. Using genome-wide KO approaches, the authors identify components that abrogate Ceapin-mediated UPR inhibition, identifying the peroxisomal ABC transporter, ABCD3, as a primary target. Through a subsequent set of thorough experiments, the authors establish that Ceapin triggers an interaction between ATF6 and ABCD3, which tethers peroxisomes to the ER and precludes traffic of AFT6 to the Golgi, presumably by steric effects. The conclusions are well-supported and uncover a novel mechanism of action for a UPR modulator, as well as suggest a new anchor-away method that will be of interest to the broader research community.

Essential revisions:

Reviewers had significant enthusiasm for the study, and overall the experiments were convincing and the conclusions largely well-supported. There are several important points raised by reviewers that don't change the conclusions of the paper, but that we agree are important to address before publication. Of particular importance is the question of the foci that ATF6 becomes anchored to upon Ceapin treatment – given that some peroxisomal membrane proteins are first inserted into the ER, it seems a formal possibility that the foci are pre-peroxisomal vesicles rather than peroxisomes. Although this doesn't change the conclusion of how Ceapin impacts ATF6 ER retention, it does affect the specific model invoked and thus is an important question. Related questions about stoichiometry and ATF6 activation are also important and speak to the model.

Thus essential revisions include: (i) address the potential question of whether ABCD3 recruits ATF6 to a bona fide peroxisome, or whether an ER subdomain or pre-peroxisomal vesicle might be the anchor; (ii) address the related question of stoichiometry to determine whether sufficient ABCD3 is expressed to quantitatively retain ATF6; (iii) check activation of ATF6 directly by immunoblot rather than reporter assay; (iv) address the questions about other candidates that were not explored further.

*Reviewer #1:*

My only major point would be that the authors don't formally demonstrate that Golgi localization and/or processing of ATF6 is restored in the ABCD3 KO + stress + Ceapin condition. It seems to be assumed that traffic and processing is restored, but this is not shown.

*Reviewer #2:*

This is a very interesting piece of work. The data presented are comprehensive and greatly improve our understanding of how Ceapins specifically inhibit ATF6α. The manuscript is well-written, clear, and easy to understand. I recommend publication of this manuscript in *eLife* after considering one major point that could have a major impact on the validity of the exciting model for Ceapin action that the authors propose (induction of a neomorphic inter-organelle tether).

Specifically, from my understanding of the literature there is a difference of opinion regarding how peroxisomal membrane proteins (PMPs such as PMP70, also known as ABCD3) end up in the peroxisome: (1) PMPs are synthesized in the cytosol and recruited into peroxisomes or (2) PMPs are first inserted into the ER membrane and then the PMP-containing pre-peroxisomal vesicle is pinched off from ER membrane. The issue of alternative models for peroxisome biogenesis is discussed in detail in Smith et al., 2013.

The authors make great use of the first option to propose a model for Ceapin-induced ATF6α inhibition (Figure 7). However, my understanding is that it is debatable whether ABCD3 is inserted directly into the peroxisomal membrane from the cytosol (as mentioned in subsection “ABCD3 is required for Ceapin-induced ATF6α foci” of this manuscript). Therefore, I am currently not convinced by the reasoning of the authors in subsection “ABCD3 is required for Ceapin-induced ATF6α foci”, and thus am not convinced by the proposed model in Figure 7 or, for that matter, that the manuscript title is appropriate. In fact, if I give credence to the model that PMPs are first inserted into ER membrane and then the PMP-containing pre-peroxisomal vesicles are pinched off from ER membrane, I could propose another model for Ceapin action that I believe is fully consistent with all the data provided in this manuscript (including work on the cytosolic domain of ATF6α). That is, Ceapin initiates the recruitment of ATF6α to the ER-derived pre-peroxisomal vesicle through ABCD3.

Citations that motivate my suggestion of the alternative model for Ceapin function are the following: Although commonly used as a reliable marker for peroxisomes, there appears to be evidence that PMP70 (or ABCD3) also presents in specialized subdomains of the ER that are continuous with a peroxisomal reticulum from which mature peroxisomes arose (Geuze et al., 2003; Smith et al., 2013. Similarly, PEX14 (found to be colocalized with ATF6α in the presence of Ceapin) is also found in these pre-peroxisomal vesicles (Joshi et al., 2016; Joshi et al., 2018). If my interpretation of the literature is right, it could provide an alternative explanation for how Ceapin induces the formation of ATF6α foci. In addition, if ATF6α could partially be translocated to the peroxisomal vesicle, it would explain the reduced activity of ATF6α in the presence of Ceapin.

I believe that microscopy methods such as conventional immunofluorescent imaging (probing for ER-resident proteins, ATF6s, and ABCD3 or PEX14 (or PEX19?) or immunolabelled electron microscopy could answer whether or not the proposed model in Figure 7 is valid or whether the alternative explanation mentioned above could be relevant. If the alternative model cannot be falsified, at a minimum the title and abstract of the paper should change to account for this fact and the Discussion section should address these issues.

*Reviewer #3:*

1) According to the presented data, ATF6α is fixed to the ER membrane due to the interaction with ABCD3 mediated by Ceapin-A7. There is no description about the molecular number of these proteins in the cell. Is ABCD3 quantitatively expressed enough numbers more than ATF6α in the cell? I am very concerned with the stoichiometry of these molecules. In IF studies (ex. Figure 2D), green colored ATF6α looks more expressed than ABCD3.

2) Although the authors obtained 20 candidates, they focused on ABCD3 and PEX19 without further explanation. There is no description about the other 18 molecules. If possible, the authors list up them or should describe why the authors abort the further analysis. Are there all false-positive?

3) Figure 1D-G, Figure 2—figure supplement 1A, B: ATF6α activation process was mostly checked by reporter assay or IF, but the authors should also present IB data about the important experiments.

4) In most of the IF data, the merge of each foci was judged by color alteration. In some figures, it is quite hard to detect the coincidence (Ex. Figure 5 ABCD3 mutants). In such a case, the authors should need more explanation in the legend. In Figure 2D, there is no description about arrowheads. In Figure 2—figure supplement 1B, the authors forgot the description (compared with Figure 1E-G).

5) The authors should describe the efficiency of KD. If possible, KD experiments should be confirmed by the recovery of original phenotype by the expression of target gene.

---

## [Author Response]

Essential revisions:Reviewers had significant enthusiasm for the study, and overall the experiments were convincing and the conclusions largely well-supported. There are several important points raised by reviewers that don't change the conclusions of the paper, but that we agree are important to address before publication. Of particular importance is the question of the foci that ATF6 becomes anchored to upon Ceapin treatment – given that some peroxisomal membrane proteins are first inserted into the ER, it seems a formal possibility that the foci are pre-peroxisomal vesicles rather than peroxisomes. Although this doesn't change the conclusion of how Ceapin impacts ATF6 ER retention, it does affect the specific model invoked and thus is an important question. Related questions about stoichiometry and ATF6 activation are also important and speak to the model.Thus, essential revisions include:i) address the potential question of whether ABCD3 recruits ATF6 to a bona fide peroxisome, or whether an ER subdomain or pre-peroxisomal vesicle might be the anchor;

In order to more directly address the whether ABCD3 recruits ATF6α to a bona fide peroxisome, we have included colocalization studies with the matrix protein Thiolase as a marker for import competent mature peroxisomes. In the presence of Ceapin, ATF6α is tethered to Thiolase containing mature peroxisomes. We measured this colocalization with full-length ATF6α in 293 cells (Figure 2—figure supplement 2) and the minimal cytosolic ATF6α(2-90) in U2OS cells (Figure 5—figure supplement 1).

ii) address the related question of stoichiometry to determine whether sufficient ABCD3 is expressed to quantitatively retain ATF6;

ABCD3 is more highly expressed than ATF6α, by approximately 10-fold. In the HeLa Proteome (Hein et al., 2015), the protein copy numbers of ABCD3 and ATF6α are 131,689 and 9,244, respectively. We previously mentioned in the discussion that ABCD3 is abundant but now added the relative abundances and above citation to the discussion.

iii) check activation of ATF6 directly by immunoblot rather than reporter assay;

We previously measured ATF6α-dependent upregulation of endogenous target genes, HSPA5 and HSP90B1 (Figure 1F-G) in NegCtrl and ABCD3 KD cells. This measures the most downstream step of ATF6α activation in which an active ATF6α transcription factor has been produced after trafficking to the Golgi and processing by S1P and S2P. In ABCD3 KD cells treated with ER stress + Ceapin, these target genes are still induced indicating ATF6α activation is not inhibited in ABCD3 KD cells.

We also added a supplemental figure measuring ATF6α translocation from the ER to nucleus in ABCD3 KD cells and ABCD3 KD complemented with full length ABCD3 (Figure 1—figure supplement 2). This result was consistent with Figure 1F-G where ABCD3 KD does not lead to constitutive translocation of ATF6α or inhibit translocation of ATF6α in the presence of ER stress.

iv) address the questions about other candidates that were not explored further. You might also consider some of the other minor points that reviewers raised, and thus their modified reviews are included below.

We tested additional hits from the screen, but they did not retest after individual knockdown. We did not include these negative data in the paper so we could focus on the molecular target of Ceapin validation. Our reporter only showed ~two-fold induction, but this was sufficient to screen with and identify ABCD3. Additionally, ABCD3 was the only hit in both the genetic screen and proteomic analysis.

Reviewer #1:My only major point would be that the authors don't formally demonstrate that Golgi localization and/or processing of ATF6 is restored in the ABCD3 KO + stress + Ceapin condition. It seems to be assumed that traffic and processing is restored, but this is not shown.

We have addressed this point above in (iii).

Reviewer #2:[…] Citations that motivate my suggestion of the alternative model for Ceapin function are the following: Although commonly used as a reliable marker for peroxisomes, there appears to be evidence that PMP70 (or ABCD3) also presents in specialized subdomains of the ER that are continuous with a peroxisomal reticulum from which mature peroxisomes arose (Geuze et al., 2003; Smith et al., 2013. Similarly, PEX14 (found to be colocalized with ATF6α in the presence of Ceapin) is also found in these pre-peroxisomal vesicles (Joshi et al., 2016; Joshi et al., 2018). If my interpretation of the literature is right, it could provide an alternative explanation for how Ceapin induces the formation of ATF6α foci.

As described by the reviewer, there are two models for peroxisomal membrane protein (PMP) insertion into peroxisomes (directly from the cytosol into peroxisomes or via the ER). We focus on ABCD3 and do not generalize to all PMPs since there is detailed work characterizing the trafficking of newly synthesized ABCD3.

To answer the question posed of whether ABCD3 is directly inserted into the peroxisome, we have added an additional citation (Imanaka et al., 1996), showing ABCD3 inserts directly into peroxisomes in vivo and in vitro. In their in vitro reconstitution system they show ABCD3 is inserted into peroxisomes and not mitochondria. The peroxisomal membrane protein targeting signal (mPTS) for ABCD3 has also been defined (Kashiwayama et al., 2007). We previously cited proximity specific ribosome profiling experiments in HEK293 cells where proteins co-translationally translocated into the ER were globally identified. We have added a new supplemental figure (Figure 2—figure supplement 1) with these data (Jan et al., 2014), indicating ABCD3 is not co-translationally translocated into the ER. As a positive control we highlight PEX16, which has been shown to be co-translationally translocated into the ER in mammalian cells (Smith et al., 2013) and is co-translationally translocated in this dataset.

In (Stroobants et al., 2001, Geuze et al., 2013), they propose a model where peroxisome biogenesis begins at specialized ER sites that separate into a reticular lamella and then transition into globular peroxisomes. In this model, the peroxisomal membrane is largely ER-derived and at a certain maturation state the functional import machinery for matrix proteins is assembled.

In order to address more directly the potential question of whether ABCD3 recruits ATF6α to a bona fide peroxisome, we have included Thiolase, a matrix protein that is a marker for import competent mature peroxisomes, as additional control. In the presence of Ceapin, ATF6α is tethered to Thiolase containing mature peroxisomes. We measured this colocalization with full-length ATF6α in 293 cells (Figure 2—figure supplement 2) and the minimal cytosolic ATF6α(2-90) in U2OS cells (Figure 5—figure supplement 1).

In addition, if ATF6α could partially be translocated to the peroxisomal vesicle, it would explain the reduced activity of ATF6α in the presence of Ceapin.

Ceapin foci are reversible (Gallagher et al., 2016). When Ceapin is washed out, ATF6α in foci quickly redistribute into the ER. Additionally, ATF6α in Ceapin induced foci are accessible to Golgi proteases when the Golgi is collapsed into the ER with BFA treatment. These data are inconsistent with ATF6α itself being inserted or moved into another compartment.

Reviewer #3:1) According to the presented data, ATF6α is fixed to the ER membrane due to the interaction with ABCD3 mediated by Ceapin-A7. There is no description about the molecular number of these proteins in the cell. Is ABCD3 quantitatively expressed enough numbers more than ATF6α in the cell? I am very concerned with the stoichiometry of these molecules. In IF studies (ex. Figure 2D), green colored ATF6α looks more expressed than ABCD3.

We have addressed this point above in (ii). Additionally, in the IF experiments, the exposure times for ATF6 and ABCD3 are not the same (3000 ms for ATF6α and 200-300 ms for ABCD3).

2) Although the authors obtained 20 candidates, they focused on ABCD3 and PEX19 without further explanation. There is no description about the other 18 molecules. If possible, the authors list up them or should describe why the authors abort the further analysis. Are there all false-positive?

We have addressed this point above in (iv).

3) Figure 1D-G, Figure 2—figure supplement 1A, B: ATF6α activation process was mostly checked by reporter assay or IF, but the authors should also present IB data about the important experiments.

We have addressed this point above in (iii).

4) In most of the IF data, the merge of each foci was judged by color alteration. In some figures, it is quite hard to detect the coincidence (Ex. Figure 5 ABCD3 mutants). In such a case, the authors should need more explanation in the legend. In Figure 2D, there is no description about arrowheads. In Figure 2—figure supplement 1B, the authors forgot the description (compared with Figure 1E-G).

Thank you for pointing this out, we added a description of the arrowheads for Figure 2D. We have added higher resolution images and did not compress figures in the updated submission. In Figure 2—figure supplement 3B, we did not forget the description. We did not measure ATF6α target gene induction by qPCR as in Figure 1F-G, but rather measured reporter gene induction as described.

5) The authors should describe the efficiency of KD. If possible, KD experiments should be confirmed by the recovery of original phenotype by the expression of target gene.

We previously indicated ABCD3 KD efficiency in Figure 2 legend (96% KD). In Figure 5 (ABCD3 mutants), this recovery experiment was done. In these experiments, endogenous ABCD3 is knocked down and the full-length or truncated mutants ABCD3 constructs were added back. The ABCD3 KD cell line complemented with full-length ABCD3 formed foci that colocalized with ATF6 upon Ceapin treatment (Figure 5C-D). This cell line (ABCD3 KD + ABCD3 FL) was also used in Figure 6C for reciprocal immunoprecipitations and physically associated with ATF6α in the presence of Ceapin.